# A gene prioritization method based on a swine multi-omics knowledgebase and a deep learning model

Yuhua Fu [1,2], Jingya Xu[1], Zhenshuang Tang[1], Lu Wang[1], Dong Yin[1], Yu Fan[1], Dongdong Zhang [2], Fei Deng[2], Yanping Zhang[2], Haohao Zhang[2], Haiyan Wang[1], Wenhui Xing[2], Lilin Yin [1], Shilin Zhu[1], Mengjin Zhu[1], Mei Yu[1], Xinyun Li[1], Xiaolei Liu[1✉], Xiaohui Yuan [2✉] & Shuhong Zhao [1✉]

The analyses of multi-omics data have revealed candidate genes for objective traits. However, they are integrated poorly, especially in non-model organisms, and they pose a great challenge for prioritizing candidate genes for follow-up experimental verification. Here, we present a general convolutional neural network model that integrates multi-omics information to prioritize the candidate genes of objective traits. By applying this model to *Sus scrofa*, which is a non-model organism, but one of the most important livestock animals, the model precision was 72.9%, recall 73.5%, and F1-Measure 73.4%, demonstrating a good prediction performance compared with previous studies in *Arabidopsis thaliana* and *Oryza sativa*. Additionally, to facilitate the use of the model, we present ISwine (http://iswine.iomics.pro/), which is an online comprehensive knowledgebase in which we incorporated almost all the published swine multi-omics data. Overall, the results suggest that the deep learning strategy will greatly facilitate analyses of multi-omics integration in the future.

[1] Key Laboratory of Agricultural Animal Genetics, Breeding and Reproduction, Ministry of Education, Key Laboratory of Swine Genetics and Breeding, Ministry of Agriculture, College of Animal Science and Technology, Huazhong Agricultural University, 430070 Wuhan, Hubei, P.R. China. [2] School of Computer Science and Technology, Wuhan University of Technology, 430070 Wuhan, Hubei, P.R. China. ✉email: xiaoleiliu@mail.hzau.edu.cn; yuanxiaohui@whut.edu.cn; shzhao@mail.hzau.edu.cn

n the past few decades, with advanced sequencing technologies, huge volumes of omics data have been produced and used to interpret the underlying genetic mechanisms of biological processes. For example, genome resequencing data were used to study the domestication history of species[1–3]; RNA-seq data were used to locate functional genes for specific tissues[4–6]; and metagenomics data helped to interpret the molecular pathways that underlie the interactions between organisms and the microbial environment[7–9]. In addition, proteomics[10], metabolomes[11], methylation[12], miRNA[13], and other omics data have all participated in our understanding of the mechanisms of various biological processes[14]. However, analyses have always been conducted on one type of omics data, such as a genome-wide association study (GWAS), which only utilize genomic information to generate a list that contains dozens or even hundreds of candidate genes, but these analyses always stop at the "association" level[15]. Even though there are massive amounts of multi-omics data, it is a challenge to integrate the information to identify further the credible candidate genes. We face the problem of having too much data, but too little knowledge.

Recent research has integrated information from multiple omics to reduce the false positives caused by studies of single omics and to improve the probability of identifying credible candidate genes[5,14,16–20]. There are two commonly used strategies to integrate information from multiple omics. One is to narrow the large set of candidate genes by selecting the overlapping regions based on evidence from different layers of multiple omics, such as selecting the candidate genes that significantly affect the objective traits in both genomics and transcriptomics[21,22]. The other strategy is to map credible candidates by constructing a network to interpret its functions and biological meanings, such as pathway analysis and co-expression network analysis, to locate the genes within the pathways that are associated with objective traits[23,24]. However, limited by sample size and experimental design, it is very rare to obtain the evidence from multiple omics information for a specific candidate gene in a single experiment. Developing a new method to make efficient reuse of omics data is urgent and essential.

Because multiple omics data are heterogeneous, it is difficult to incorporate multiple omics information in a classic statistical model. The machine learning method has proved to be a powerful tool to handle a large amount of heterogeneous information, and it has great potential to solve the problem of multiple omics integration[25]. In fact, machine learning methods have already been widely used in biological research, such as clinical studies[26,27], disease risk assessment[28,29], genomic prediction[30–32], and mining of biological literature[33,34]. Deep learning, which is a branch of machine learning, is helpful in solving the problem of extracting target features using traditional machine learning methods. Deep learning can learn the intricate regulatory relationships among the diversified multi-omics and, at the same time, it has excellent potential to integrate the features of multi-omics information. Recently, deep learning methods have been used successfully to assist the diagnosis and assessment of disease risk by integrating features of multiple omics[35–37]. However, its application is limited to specific samples or scenarios and to reuse multi-omics data is still challenging. Additionally, the poor interpretability of deep learning models also perplexes researchers[38]. At the moment, text mining technology, which is used widely to extract knowledge of relationships between genes and traits from published literature, brings a ray of hope for solving this problem. To incorporate this information into the multiple omics integration model not only improves the interpretation of results, but it also saves time when searching a vast amount of literature manually.

Obviously, multi-omics analysis depends on advanced integration methods and a large amount of regular multi-omics information. Most existing integration methods are shared in the form of source code only, where multi-omics data are not included. To provide the integration method with an easy-to-use multi-omics database would benefit data reuse and model sharing.

With these design criteria in mind, we constructed a multi-omics database for swine, which is an important non-model organism and one of the most important livestock animals. The database contains almost all the published pig genome data, transcriptome data, and QTX data. Using these data, we trained the integration model based on a machine learning strategy to prioritize candidate genes for objective traits and to evaluate the biological significance of the model parameters. The convolutional neural network model had the highest model accuracy and an in-depth understanding of complex biological processes. In addition, to facilitate the use of the integration model and multi-omics information, we developed a user-friendly, online website named ISwine for multi-omics integration analysis and data search. With this framework, users can use multi-omics information easily to prioritize the genes of interest for a specific trait. ISwine will serve as a knowledgebase for swine research and potentially provide a new strategy for multi-omics integration analysis to benefit research on other species.

## Results

**Omics data collection and management.** ISwine collected the public multi-omics data of swine from 305 projects and 653 published sources (Table 1). All the resequencing data from 42 projects were included in this study, which contained 864 pig individuals (Supplementary Table 1). A total of 32.88 terabytes (TB) of resequencing data were aligned against the Sscrofa11.1 reference sequence by using BWA (Burrows–Wheeler Aligner) software[39]. To call variants with high confidence, potential duplications were filtered for each individual, and individuals with <3-fold depth and 70% genome coverage were removed (Supplementary Fig. 1 and Supplementary Data 1). A total of 825 qualified individuals were retained for subsequent analyses, which included 29 Asian native breeds, 20 European native breeds, three European commercial breeds, two American native breeds, and five other breeds (Supplementary Fig. 2 and Supplementary Table 2).

After applying stringent quality control criteria (details described in "Methods"), a total of 81,814,111 SNPs and 11,920,965 indels were identified, of which 51.4 million were intergenic, 35.8 million were intronic, and 1.2 million were exonic (Supplementary Table 3). Compared with the variants that were included in the pig dbSNP (Build 150) database[40], our variant data set covered >74.02% of its variants, and 46,451,715 variants were considered as novel because they were not in the pig dbSNP (Supplementary Fig. 3). These novel SNPs will improve the catalog of porcine genetic variants substantially, and the high coverage of dbSNP reflected that the genomic data set in this

**Table 1 Overview of the multi-omics data used to construct the integrated swine omics knowledgebase.**

| Omics | Samples | Classification | Scale | Source |
|---|---|---|---|---|
| Genome | 864 | 59 breeds | 32.88 TB | 42 projects |
| Transcriptome | 3526 | 95 tissues | 20.0 TB | 263 projects |
| QTXs | – | 89 traits | 26,357 entries | 653 studies |

study is a reliable reference for porcine research. This is especially true for cases where the NCBI has not maintained data on swine dbSNP since 19 April 2018.

Consistent with previous studies[41], the population genetic structural analysis showed a clear evolutionary split and introgression between Asian and European pigs (Supplementary Figs. 4 and 5). The neighbor-joining tree also showed this divergence where the Asian and European pigs defined their own separate clades, and each clade split into a domesticated clade and a wild clade, respectively (Supplementary Fig. 6). It was very clear that both Asian domestic pigs and wild boars were divided into a southern clade and a northern clade, respectively, and interestingly, the European Gottingen Minipig showed more genomic similarity to Asian southern pigs than to European native breeds (Supplementary Fig. 6). Overall, the structure analysis suggested that the collected genomic data were of high quality and covered the pig genomic diversity quite well.

The swine transcriptome data, which consisted of 20.0 TB of sequencing data, 263 projects, and 3526 samples (Supplementary Data 2), were downloaded and analyzed. A strict quality control procedure was carried out to ensure the validity of the data. First, we removed the samples with mapped reads <6M (Supplementary Fig. 7), and second, we deleted the samples with dubious tissue information based on the Euclidean distance of the samples (Supplementary Fig. 8). After deleting 244 samples by the two steps, the remaining 3282 samples were mainly grouped together in the tissue classification (Supplementary Data 3). However, the degree of dispersion varied among different tissues, which may have been related to the temporal and spatial specificity of the tissues or the sample collection method (Supplementary Fig. 9). To facilitate the retrieval of sample information, we classified all samples into seven main categories and 95 subcategories based on tissue classification and their relative position in the swine's body (Supplementary Data 4). After statistical analysis, we found that the samples were mainly concentrated in the blood and longissimus dorsi muscle categories, which was consistent with the areas of pig research that focused primarily on a disease model and meat production. In addition, the liver, endometrium, back fat, and heart tissues also had a sample size >100 (Supplementary Data 4 and Supplementary Fig. 10), which provided a high-quality reference dataset for gene expression to train the multi-omics integration model.

To better interpret the omics data, we obtained 8371 quantitative trait-associated loci (QTALs), 1,542 quantitative trait-associated genes (QTAGs), and 16,444 quantitative trait-associated nucleotides (QTANs) from 653 studies (Supplementary Data 5) by using a text mining technique[42] combined with manual markup; these information are referred to as QTXs. After mapping all reported QTXs to the pig genome Sscrofa11.1, a total of 24,238 QTXs were retained, which covered 96.6% of the entire genome (Supplementary Data 6 and Supplementary Fig. 11). Because some extremely long QTALs were detected in time with low marker densities, we cut the QTALs that were longer than two mega base pairs (MB) to 2 MB along the center of the QTAL. The retained QTXs covered 72.01% of the entire genome and 74.59% of the total genes, which was consistent with the general consensus that most genes were involved in various life activities. We selected gene sets with QTXs that were reported >30 times (Supplementary Data 7, Supplementary Fig. 12) and found that the genes focused mainly on disease and development-related functions (Supplementary Table 4). A trait ontology was built to facilitate the construction of connections between omics information and trait information. The traits were divided into 11 major categories and 89 subcategories (Supplementary Data 8). QTXs were concentrated mainly in the "Fat Related Traits", "Blood Related Traits", and "Meat Quality Traits" categories, which was

consistent with the mainstream research on pigs (e.g., production, disease, and reproduction) (Supplementary Fig. 13 and Supplementary Table 5).

**Construction of gene prioritization model.** A total of 842 samples was prepared for training the gene prioritization models, which consisted of 421 pairs of all QTAGs and relevant traits as positive samples and randomly assigned 421 pairs of genes and traits as negative samples (Supplementary Data 9). Then, 80% of the samples were used as the training set, and the remainder were used as the test set. To make full use of multi-omics data, the features of variation counts, expression levels, QTALs/QTANs number, and the WGCNA (Weighted Gene Co-expression Network Analysis) module for each gene of the training dataset were obtained for integration analysis. Using these features, four models were trained to prioritize the candidate genes obtained from GWAS or any other omics analysis that was associated with a trait of interest. This included two machine learning models with great interpretation capacity, logistic regression classifier (LR), and linear support vector classifier (linearSVC), which are linear models that are used normally in binary classification problems. The two additional models were two neural network-based deep learning models, multi-layer perceptron (MLP) and convolutional neural networks (CNN), which have the potential to mine the interactions among the features. The candidate genes with a probability >50% were denoted as credible candidate genes (Fig. 1). For each model, a 4-fold cross-validation procedure was conducted to evaluate the model's performance, and this suggested that the CNN model performed best. The accuracy, precision, recall, and F1-Measure of the CNN model were all >70%. However, this is not particularly high for the binary classification problem, but it has great application value in biological research when the false positive rate is as low as 30% (Table 2 and Supplementary Fig. 14).

The lime framework (https://github.com/marcotcr/lime) was conducted further to understand the working principle of the CNN model, which suggested that "Nonsynonymous_indel", "Intron_snp", "Expression", "Module", and "QTAL" played important roles in gene prioritization (Fig. 2a and Supplementary Table 6). Non-synonymous variations affected gene function by changing protein coding and gene expression level, expression module determined the occurrence and intensity of gene function, and QTAL information from published sources provided direct evidence for gene function. These features were consistent with the rules of artificial gene function judgement, which meant that the model had strong interpretability and indicated that the gene prioritization model held strong biological meaning.

To further confirm the role of each omics in the gene prioritization model, we trained the LR, LinearSVC, MLP, and CNN models using the features of genomes, transcriptomes, and data from the literature. We found that among these models, the performances of models that trained by multi-omics information were better than that of single omics, and the methods based on neural network were superior to the linear methods (Supplementary Table 7). Interestingly, the performance of the model constructed using only genomic features was the best, followed by the model that used transcriptomes, and the model based on the literature was the worst. This was consistent with the evaluation results of lime framework, which indicated that the evaluation results of the integrated model were reliable. In addition, the performance of the model trained only with genomic features was closer to the integrated model, and this may be because the genomic features accounted for 71.43% (10/14) of all features. However, from a biological point of view, the information based

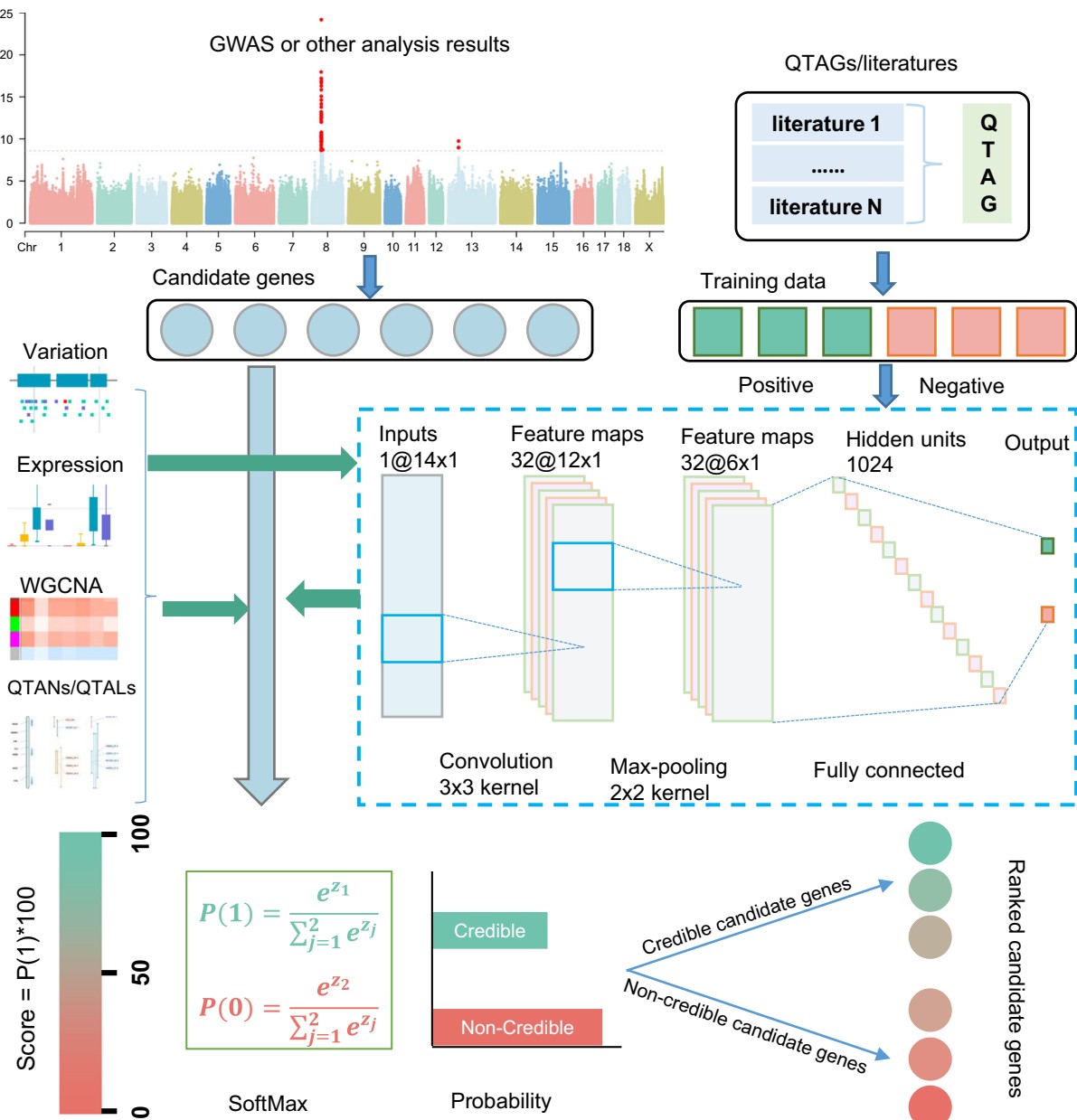

**Fig. 1 Schematic of the gene prioritization framework for the integrated swine omics knowledgebase.** The circles represent a list of candidate genes from GWAS or any other omics analysis. The rectangles represent positive training samples and negative training samples. The dotted box represents a CNN model trained by using variation counts, expression level, QTANs/QTALs number, and WGCNA module features of the training data. The output layer of the model shows the probability that the gene is a credible candidate gene by using the "softmax" function. The candidate genes with a probability >50% were denoted as credible candidate genes and can be ranked according to their probability.

**Table 2 Comparison of the performance of the four machine learning models for gene prioritization.**

| Models | Accuracy | Precision | Recall | F1-Measure |
|---|---|---|---|---|
| LR | 0.657 | 0.686 | 0.571 | 0.623 |
| LinearSVC | 0.639 | 0.658 | 0.571 | 0.612 |
| MLP | 0.692 | 0.678 | 0.726 | 0.701 |
| CNN | 0.734 | 0.729 | 0.738 | 0.734 |

The F1- Measure of the two linear models (LR and LinearSVC) with strong explanatory power were lower than deep learning models (MLP and CNN) that were based on neural networks, which suggested the deep learning models were superior to the linear models. Between MLP and CNN, the accuracy, precision, and F1- Measure of CNN were higher than MLP, and the performance of CNN was slightly better than MLP.
*LR* Logistic regression, *LinearSVC* Linear Support Vector Classifier, *MLP* Multi-Layer Perceptron, *CNN* Convolutional Neural Networks.

on transcriptomes and the literature were more interpretable. All the transcriptome features and literature features were significantly different in the positive and negative samples (Supplementary Table 8 and Supplementary Fig. 15), which also confirmed the interpretability; this suggested that some unknown regulatory mechanism in the genomic layer needed to be mined. The integration of a number of features from multiple omics not only improved the performance of the prediction model, but it also enhanced the interpretability of the model.

**Evaluation of gene prioritization model.** A total of nine traits from seven GWAS studies[43–49] were selected for testing the performance of the gene-prioritization model (Supplementary Table 9). All candidate genes were evaluated by the CNN model,

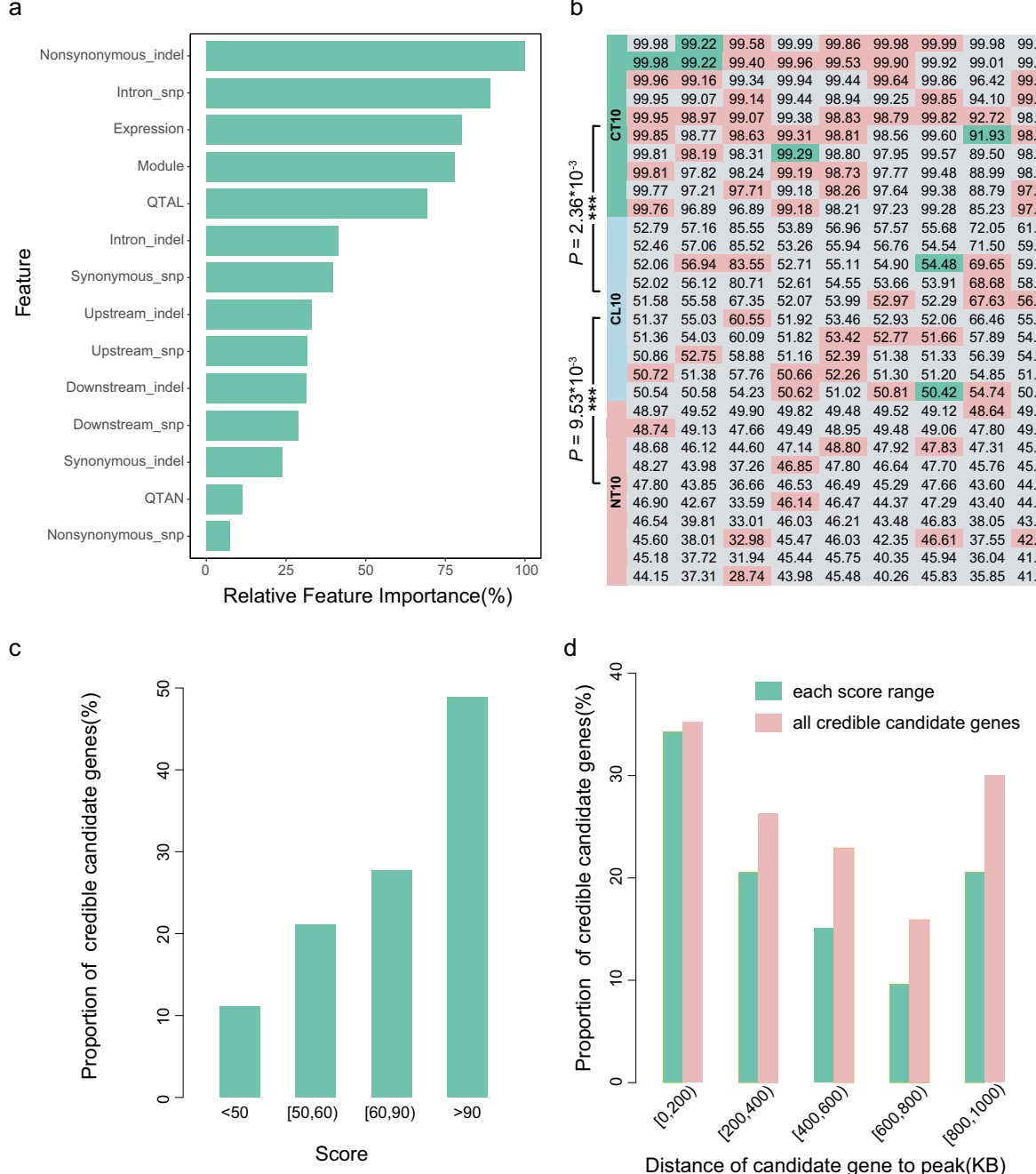

**Fig. 2 Evaluation of the CNN model. a** The relative importance of 14 features used in the CNN model. Except for the top five features, the relative importance of other features was <50%, and the top five features may have played important roles in gene prioritization. **b** The scores of candidate genes in nine real case studies. Each column represents one real case study, and each grid represents the score of a candidate gene. The green, pink, and gray backgrounds represent that the candidate gene was reported to be a credible candidate in the case literature, in other published sources, or non-reported, respectively. The CT10 and CL10 means top 10, last 10 genes from the predicted credible candidate genes, and the NT10 means top 10 genes from predicted non-credible candidate genes. **c** Proportion of credible candidate genes identified in different score ranges; genes with higher scores were more likely to be credible candidate genes. **d** Proportion of credible candidate genes relative to the distance of credible candidate gene from the peak. Candidate genes close to the peak have a higher proportion of credible candidate genes than those far away from the peak, but the proportion of credible candidate genes in near and far distance ranges were similar, which indicated that distal regulation should be considered in the identification of credible candidate genes.

and the candidate genes with a score over or under 50 were designed as "credible" or "non-credible", respectively. Overall, 50.41–82.58% of the candidate genes were predicted to be non-credible candidate genes, which greatly narrowed the scope of credible candidate genes (Supplementary Table 9). To assess the prioritizing effect further, we selected the top 10 genes (CT10)

from predicted credible candidate genes, the last 10 (CL10) genes from predicted credible candidate genes, and the top 10 (NT10) genes from predicted non-credible candidate genes for each trait for validation by consulting the literature. The number of credible candidate genes in CT10 was much greater than CL10 ($P = 2.36 \times 10^{-3}$), and the credible candidate gene number in CL10

was greater than NT10 ($P = 9.53 \times 10^{-3}$), which indicated that the recommendation of credible candidate genes by the CNN model was reliable (Fig. 2b and Supplementary Table 10). The statistical results indicated that the proportion of credible candidate genes in different scoring ranges increased with the gene score, which meant that a candidate gene was reliable if its gene score was high enough (Fig. 2c, Supplementary Table 11).

The feature differences of candidate genes in different scoring ranges were also compared to reveal the working principles of the CNN model. Nine of the 14 features, which included Module, Expression, Intron_snp, Synonymous_snp, Nonsynonymous_snp, Upstream_indel, Intron_indel, Synonymous_indel, and Nonsynonymous_indel showed significant differences among three groups (CT10, CL10, and NT10) of genes (Supplementary Table 12 and Supplementary Fig. 16). However, except for the features of Intron_snp, Intron_indel, Expression, and Module, the trends of the other five features neither increased nor decreased in CT10, CL10, and NT10, which indicated that these features were utilized in a nonlinear way. Although there were no significant differences in features of QTAL and QTAN, both of them exhibited changing trends within the three groups, which indicated that they may be potentially correlated to credible candidate genes; this was consistent with our general perception. Interestingly, a large proportion of credible candidate genes was located far away from the GWAS peaks, which indicated that distal regulation should be considered in the identification of credible candidate genes (Fig. 2d and Supplementary Table 13).

**ISwine: an integrated swine omics knowledgebase**. To facilitate model usage, an integrated swine omics knowledgebase named ISwine was constructed to prioritize candidate genes based on multi-omics information. ISwine consists of an integrated swine omics database and a computing framework for gene prioritization. The integrated swine omics database is composed mainly of three basic databases on genome variation, gene expression, and QTX and a multi-omics integration database, which uses data on multi-omics for gene prioritization and to provide information about genes (e.g., sequences and annotation). ISwine further provides three tools for users to mine or to visualize multi-omics data, which include BLAST, Primer, and JBrowser (Fig. 3 and Supplementary Fig. 17).

Three basic databases provide information, as comprehensively as possible, on swine genome variation, gene expression, and QTXs. The variation database consists of genotypes of 825 individuals at 93.7 million variations, the frequencies and annotations of all variations, and individualized information (e.g., name, sex, geographic location, and breed) (Fig. 3a, b). The expression database provides the expression of 25,880 genes with 3282 samples, the sample properties of tissues, treatment, preservation, etc., and differential expression genes (Fig. 3c, d). The QTX database includes genome coordinate positions, variation types, sources, associated traits, and trait ontology for 24,238 QTXs, and a user rating system is also provided to improve the reliability of QTX information (Fig. 3e, f).

The integrated database not only provides the call interface to basic databases, but also contains the basic information, sequences, annotation, and homologous genes for all genes, which can help users to better judge the gene function (Fig. 3g, h). Combined with the CNN model embedded in the integration database, by providing the physical positions/gene IDs for genes of interest and selecting the objective trait and the tissues potentially related to the trait, users can easily obtain the priority and scores of candidate genes to identify credible candidate genes (Fig. 3i). Users can further use the three tools implemented in ISwine to perform downstream analyses (e.g., using Primer to

design primers for subsequent experiments (Fig. 3j), using JBrowser to visualize the genome (Fig. 3k), and BLAST to achieve the target gene sequences (Fig. 3l). To facilitate the usage of databases, tools, and methods further, ISwine provides a user-friendly interface to browse, search, visualize, download, and analyze multi-omics information.

## Discussion

Mining the information of multi-omics can help us to understand the biomechanisms that underlie agricultural economic traits and human diseases. However, there are always two challenges: one challenge is to collect and organize the multi-omics information, which are massive and have a lack of standardization, and the other challenge is how to integrate heterogeneous multi-omics information. Our study provides a new strategy to integrate the massive quantity of multi-omics information, a knowledgebase, which not only supplies various user interactive query functions for browsing, searching, visualizing, and downloading multi-omics data, but also provides a machine learning strategy-based gene prioritization method to integrate multi-omics information. As we know, GWAS is one of the most widely used methods to identify candidate genes that underlie traits. However, it is always a big challenge to rank credible candidate genes after association tests, and our strategy may help researchers to get through the "last mile" of GWAS.

ISwine is the first integrated multi-omics knowledgebase for swine, and it is mainly composed of three basic databases titled Variation, Expression, and QTX, and an integrated database with an embedded gene prioritization method. The present version of ISwine consists of almost all the published swine data, which include 81,814,111 SNPs and 11,920,965 indels from 825 resequencing individuals (32.9 TB), 25,880 genes from 3,282 RNA-seq samples (20.0 TB), and 24,238 QTXs from 586 published sources. For the database Variation, compared with existing swine databases, such as pigVar[50], dbSNP from NCBI (updates have stopped), and the Genome Variation Map[51], ISwine has the largest number of variations and the largest number of resequencing individuals (Supplementary Table 14). The database Expression is the first RNA-Seq based, swine gene expression database, which incorporates data from 263 studies and a total of 95 various tissues. pigQTLdb[52] is the only existing database that provides information similar to the ISwine QTX database. ISwine is different from pigQTLdb because it not only collected QTX information, but it also focused on improving its interactivity with the databases of other omics.

Abundant omics sequencing technology has greatly promoted biological research and brought great challenges to the integration of various omics data. ISwine provides an online gene prioritization platform based on multi-omics data, which can directly help users obtain the rank of candidate genes for a trait of interest. A machine learning model is trained by using the features of variation counts, expression level, QTX number, and a WGCNA module in the databases Variation, Expression, and QTX, and it aims to identify the candidate genes with high confidence. The 4-fold cross-validation results showed that the accuracy, precision, recall, and F1-Measure of the CNN model were all >73%. Compared with the recall of 64% and 79% in *Arabidopsis thaliana* and *Oryza sativa*, respectively[53], the prediction performance of our model was good. Although the recall in *Oryza sativa* was higher than in our study, the genes were only prioritized in the QTL regions, while we prioritized genes in the whole genome. However, there still existed ~27% false positive/negative results in our study; one possible reason may have been the quality of the training dataset, which affected the performance of the gene prioritization model. In an ideal situation, all genes in

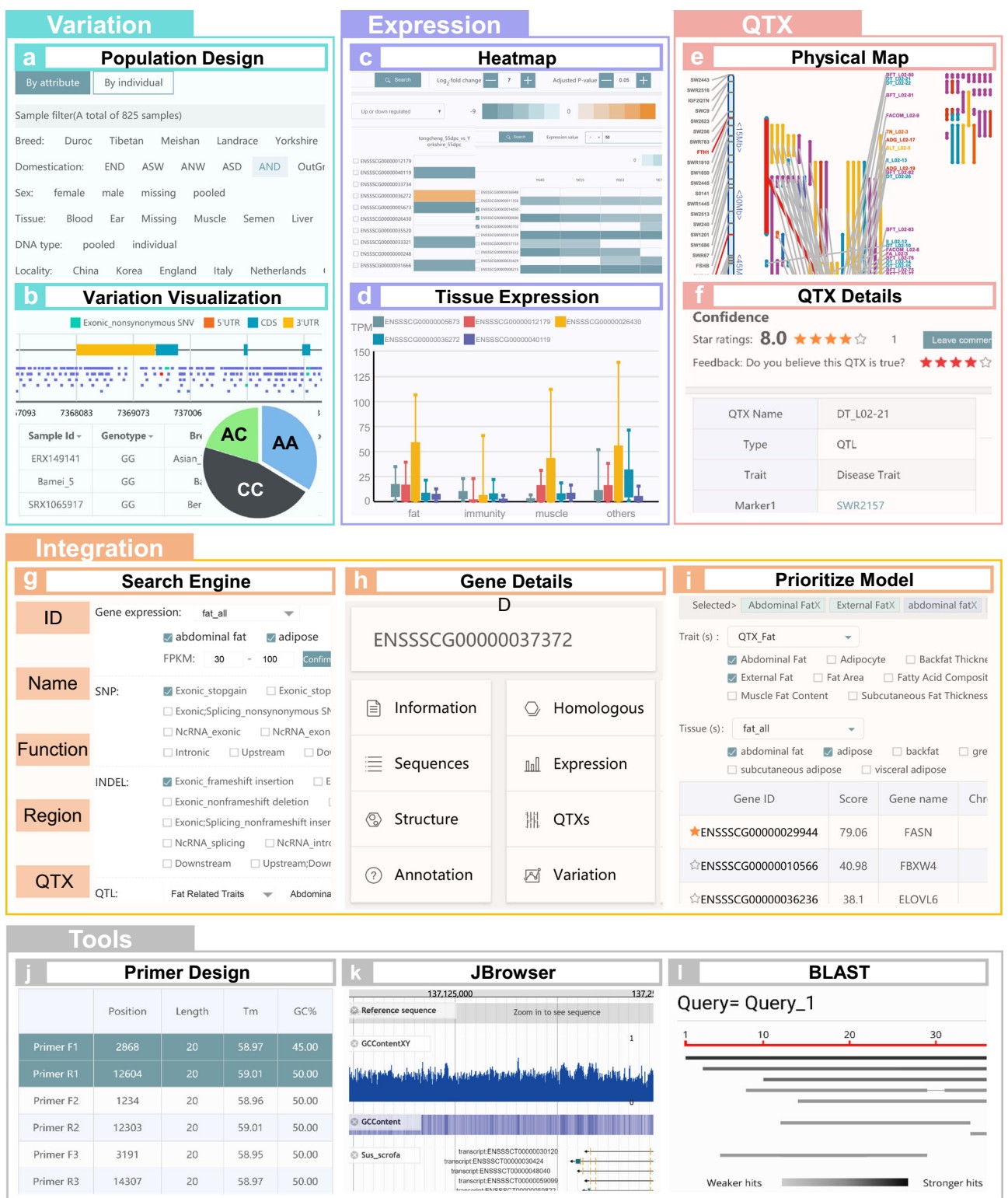

**Fig. 3 Interface and general functions of the integrated swine omics knowledgebase. a** The population design module in variation database, **b** the variation visualization module in variation database, **c** the Heatmap page in expression database, **d** the tissue expression module in expression database, **e** the physical map in QTX database, **f** the QTX information page in QTX database, **g** search engine of the integration database, **h** the gene information incorporated into the integration database, **i** the gene prioritization model embedded in integration database, **j** the primer design module for primer design, **k** the JBrowser module for genome visualization, and **l** the BLAST module for location of target gene sequences.

positive cases should influence the trait either directly or indirectly, and all negative genes should be completely unrelated to the objective traits. However, in practice, due to the complexity of gene functions, the training data set is never completely accurate. Therefore, we checked all cases of positive and negative genes manually to improve the quality of the training dataset.

In this study, linear models with strong explanatory power and deep learning models based on complex neural networks were designed. Linear models were more consistent with the concept of the expert scoring system, but the deep learning models assumed that the information of multi-omics had complex interactions. The deep learning models were superior to the linear models, which indicated that there was indeed a certain degree of interaction among multi-omics, and the rules based on linear superposition only partially explained the biomechanism that underlay traits of interest. The undeniable fact is that the deep learning models were less interpretable because they were limited by their complex network structure. Therefore, we tried to interpret the CNN model with the best performance and found that the important features of the model had strong interpretability, which was consistent with the frequently used biological rules, such as non-synonymous variations and gene expressions. At the same time, the model also reflected some unexplainable features, which could be caused by the complexity of the regulatory network of life activities that cannot be explained fully by known explainable features. The results also indicated that the deep learning models had great potential to exploit new biological rules. With the explosive increase in data, the deep learning models are bound to play an important role in multi-omics integration analyses, the exploration of biological regulation mechanisms, and studies of gene-gene interaction networks.

ISwine introduced literature information into the integration analyses innovatively, which improved the accuracy of the gene prioritization model and its interpretability. Although the amount of QTX data was far less than other omics, QTX information still weighed heavily in the gene prioritization model, which means that information from the literature played an important role in multi-omics integration analyses. With the development of text mining technology, a massive quantity of information from published sources will be extracted, structured, and used for the interpretation of omics data. Therefore, the importance of published information will increase prominently, and it is a potentially important research direction in the near future.

With the rapid increase of multi-omics data, multi-omics integration methods are lacking and, therefore, integration tools are needed. At present, multi-omics integration research can only share ideas and results by sharing codes and text descriptions. Meanwhile, multi-omics data are generated based on the design of a single study, which makes the data difficult to reuse in other research. To solve the problems, we designed ISwine, which allows researchers to use the online gene prioritization system with the latest deep learning model with only a few clicks.

However, the multi-omics integration knowledgebase and the embedded gene prioritization method have only been developed for swine. It has provided a novel strategy for multi-omics integration and could be expanded to multi-omics studies in other species. In addition, our study only applied the multi-omics integration model to the identification of credible candidate genes. It could be applied to other analyses as well, for example, to provide multi-omics information based on prior knowledge for gene differential expression analysis and a GWAS model.

In summary, we present an integrated multi-omics knowledgebase with an embedded CNN model for prioritizing genes of interest by multi-omics integration analyses, and we demonstrated the great potential of CNN model in multi-omics integration by applying it to a non-model organism—swine. To facilitate the use of model and multi-omics data, we implemented the comprehensive knowledgebase in a website named ISwine. It contains abundant genomic, transcriptome, and QTX data and massive annotation resources. A gene prioritization system was designed to help users catch the credible candidate genes by integrating information from different omics layers, and a number of analysis tools, which included JBrowse, BLAST, and Primer, were implemented to help users perform the subsequent analyses. ISwine will help us to better understand the biomechanisms that underlie the traits of interest in swine, and our novel strategy will also benefit multi-omics integration research in other species.

## Methods

**Data collection**. Resequencing and RNA-seq data were downloaded from NCBI Sequence Read Archive[54] (SRA, http://www.ncbi.nlm.nih.gov/sra/), the European Nucleotide Archive[55] (ENA, https://www.ebi.ac.uk/ena), and NCBI Gene Expression Omnibus[56] (GEO, http://www.ncbi.nlm.nih.gov/geo). This study included 3,526 RNA-seq samples from 263 projects and 864 resequencing samples from 42 projects; it covered almost all the published resequencing data and RNA-seq data in swine researches.

QTX-related literature were obtained from US National Library of Medicine National Institutes of Health[57] (PubMed, https://www.ncbi.nlm.nih.gov/pubmed) by using "swine GWAS" or "swine QTL" as keywords. A total of 653 full texts was collected for QTX detection.

Information on genetic markers were obtained from ARKdb genome database[58] (http://www.thearkdb.org) and NCBI Nucleotide database (www.ncbi.nlm.nih.gov/nuccore/). Genome sequences and annotation of Sscrofa11.1 were download from the Ensembl genome browser[59] (http://www.ensembl.org).

**Analysis of RNA-seq data**. All collected datasets were processed through the following procedure. The raw data were first converted to fastq files by using SRAToolkit[54] (V2.8.2), and the fastq files were trimmed by removing adapters and low-quality bases using Trimmomatic[60] (V0.36). The clean reads were then aligned to the Sscrofa11.1 reference genome using HISAT2[61] (V2.1.0). Reads count for each gene was extracted by using HTseq[62] (V0.9.1), and only uniquely aligned reads were retained for subsequent analysis. The differentially expressed genes (DEGs) were then identified by using DEGseq[63] and DESeq2[64]. TPM (Transcripts Per Million) was calculated by using Stringtie[65] (V1.3.3b).

To detect the samples with incorrect tissue information, we calculated the Euclidean distance between each pair of samples in each tissue and detected the outliers by the Tukey's fences method. The tissue information of these abnormal samples was considered to be incorrect, and these samples were excluded from subsequent analyses. The Tukey's fences method defined any observations that were outside the following range to be outliers:

$$[Q1 - k \times (Q3 - Q1), Q3 + k \times (Q3 - Q1)]$$

where $Q1$ and $Q3$ represent the first and third quartile of Euclidean distance observations, respectively; where $k$ is a nonnegative constant, and $k = 1.5$ or $k = 3$ indicated an "outlier". $k$ was set to 3 in this study.

**Identification and annotation of short variants from resequencing data**. Similar to RNA-seq data, all collected datasets were also processed through a consistent bioinformatics pipeline. After conversion and trimming, the remaining high-quality reads were aligned against the Sscrofa11.1 reference sequence by using Burrows–Wheeler Aligner 0.7.17[39] (BWA). The uniquely aligned reads with ≤5 mismatches were used for detection of short variants.

To obtain highly confident short variants, we employed GATK[66] (V4.0.3.0) variant calling pipelines, based on the GATK best practice online documentation. The SNPs/indels were then filtered further by the "QUAL < 30.0 || QD < 2.0 || FS > 60.0 || MQ < 40.0 || SOR > 3.0 || ReadPosRankSum < −8.0" / "QUAL < 30.0 || QD < 2.0 || FS > 200.0 || SOR > 10.0 || ReadPosRankSum < −20.0 || MQ < 40.0 || MQRankSum < −12.5" option, and high-quality short variants were retained for annotation by using ANNOVAR[67] (V2018Apr16).

**Collection of QTX information from the literature**. We used the table mining method as described in our previous study[42] to identify QTX information from the literature and checked the information manually. Each QTX record had at least two attributes: a coordinate and a trait. To aggregate the information from different genetic maps or physical maps, the Sscrofa11.1 reference genome was selected as a reference physical map, and all other maps were converted to the reference physical map by using Bowtie2[68] (V2.3.4.3).

**Annotation of the Sus scrofa genes**. Basic information, gene structure, and gene sequences were extracted from the Sus scrofa genome file and the annotation file (Ensembl release 95). GO annotations and homologous information were obtained

from the Ensembl BioMart database[59] (http://www.ensembl.org/biomart). KEGG annotations were created by using KOBAS[69] (V2.1.1), and InterPro annotations were created by using InterProScan[70] (V5.27-66.0).

**Detection of co-expression modules.** Weighted correlation network analysis (WGCNA) was used to identify co-expression modules, and the R software package WGCNA[71] (V1.67) was implemented for all genes in our collected RNA-seq samples. Genes with too many missing samples or zero variance were filtered by the "goodSamplesGenes" algorithm in WGCNA for further analysis. Then, samples were used to calculate Pearson's correlation matrices, and the weighted adjacency matrix was created with the following formula:

$$a_{ij} = \left| \mathrm{cor}(x_i, x_j) \right|^{\beta}$$

where $x_i$ and $x_j$ represent gene expression of gene $i$ and gene $j$, respectively, $a_{ij}$ is the adjacency between gene $i$ and gene $j$, and $\beta$ is the soft-power threshold. Once the gene network was constructed, a topological overlap measure was used to detect gene modules. The minimal gene module size was set to 30, and the threshold to merge similar modules was set to 0.2.

**GO term and KEGG pathway enrichment analysis.** GO/KEGG enrichment analysis provided all GO terms/KEGG pathways that were significantly enriched in candidate genes compared with the genome background, and it selected the candidate genes that corresponded to biological functions. All candidate genes were first mapped to GO terms/KEGG pathways in the annotation files by calculating gene numbers for every term; then, it used a hyper geometric test to find significantly enriched GO terms/KEGG pathways in candidate genes compared with the genome background. The significance of each GO term/KEGG pathway was calculated as:

$$P = 1 - \sum_{i=0}^{m-1} \frac{\binom{M}{i}\binom{N-M}{n-i}}{\binom{N}{n}}$$

where $N$ is the number of all genes with GO/KEGG annotations, $n$ is the number of candidate genes in $N$, $M$ is the number of genes annotated for any certain GO term/KEGG pathway, and $m$ is the number of candidate genes in $M$. The calculated P-values of all GO/KEGG terms were corrected by the Bonferroni Correction method. Taking corrected P-value ≤ 0.05 as a threshold, the GO terms/KEGG pathways that fulfilled this condition were considered as significantly enriched GO terms/KEGG pathways for candidate genes.

**Training set preparation.** To create a reliable dataset to train the machine learning model is very important, however, it is difficult to determine whether a gene is related to an objective trait, especially in research on non-model organisms. In swine research, the QTALs and QTANs were always identified by the statistical analyses (e.g., QTL mapping or GWAS) only without any experimental certification. Differently, the published studies for QTAGs were always conducted with an additional experimental certification after statistical analysis, and these QTAGs were more reliable than QTALs and QTANs. Therefore, we used QTAGs as positive samples and, finally, we obtained 421 effective gene-trait pairs (Supplementary Data 10). These samples affect "Behavioral Traits", "Blood Related Traits", "Disease Related Traits", "Exterior Traits", "Fat Related Traits", "Growth Related Traits", "Meat Quality Traits", "Muscle Related Traits", "Physiochemical Traits", "Reproduction Traits", and "Slaughter Traits" (Supplementary Data 11). To generate credible data for negative samples, we randomly generated 750 hypothetical QTAGs for the above traits, and then we removed the genes that might affect the corresponding traits by consulting the literature manually. Finally, we also randomly selected 421 genes as negative samples to ensure the balance of positive samples and negative samples. Finally, a dataset that composed of 842 samples was used to train the model; we randomly selected 80% of the samples as the training set, and the remaining 20% was the test set (Supplementary Data 9).

**Feature generation.** We generated a total of 14 features to train the gene prioritization model, which included 10 genome features, two transcriptome features, and two literature features.

Genome features: at the genomic level, we used the number of SNPs/indels within or nearby the gene as a variation feature for two reasons: (1) when SNPs/indels existed within a gene or in a regulatory region near a gene, they may have played a direct role in regulating gene function and, therefore, also the trait, especially nonsynonymous variations. (2) genes that do not have variations within or near a large population always have difficulty affecting a trait. Because the effect of variation in different regions of the gene was different, the number of SNPs and indels located upstream (two kilobases upstream of the gene), downstream (two kilobases downstream of the gene), introns, and exons (including both synonymous and nonsynonymous mutations) were also recognized as genomic features.

Transcriptome features: at the transcriptome level, genes need to be expressed to perform their functions to affect traits, and the strength of gene functions is always related to the gene expression level. Therefore, the expression levels of the genes in the target tissues were treated as an expression feature, and the correlation coefficient between the co-expression module and the tissue was used as a network feature.

Literature features: considering a large number of published sources, if a gene was reported to be associated with a trait a large number of times, it was more likely to be a reliable candidate gene. Therefore, we used the number of QTALs and QTANs, which overlapped with the gene region, as literature features.

**Model training and gene prioritization.** The scikit-learn framework[72] (V0.21.3) was used to train the logistic regression classifier (LR), linear support vector classifier (linearSVC), and multi-layer perceptron (MLP) model. To choose appropriate parameters, grid searches with 4-fold cross-validation were used to estimate the best parameters for each model. The final parameters of LR were set as: "penalty='none', solver='newton-cg'"; the final parameters of linearSVC were set as: "tol=0.00001, loss='squared_hinge', C=0.1, penalty='l2'"; and for MLP, the final parameters were set as: "activation='tanh', solver='lbfgs', hidden_layer_-sizes=(8,8), tol=0.00001, max_iter=60".

The LR is a linear model, and the probabilities for credible candidate genes and non-credible candidate genes were calculated as follows:

$$P(Y = 1|x) = \frac{e^{w \cdot x + b}}{1 + e^{w \cdot x + b}}$$

$$P(Y = 0|x) = \frac{1}{1 + e^{w \cdot x + b}}$$

where $x$ is a vector of features, $w$ is the vector of feature weights, $b$ is the intercept from the linear regression equation, $Y = 1$ represents credible candidate genes, and $Y = 0$ represents non-credible candidate genes.

Similar to LR, the linearSVC is also a linear model, and a hyperplane was used to classify the credible candidate genes and non-credible candidate genes by signum, which is a classifier decision function:

$$f(x) = \mathrm{sign}(w^* \cdot x + b^*)$$

$$\mathrm{sign}(z) = \begin{cases} 1 & , z > 0 \\ 0 & , z = 0 \\ -1 & , z < 0 \end{cases}$$

where $x$ is a feature vector, $w^*$ is a normal vector to the hyperplane, $b^*$ is the intercept, and $z$ represents the $w^* \cdot x + b^*$. If the training data were linearly separable, a hyperplane, which is defined as $w^* \cdot x + b^* = 0$, was selected to separate the data into two classes represented by $f(x) = 1$ and $f(x) = -1$.

The MLP is a class of feedforward artificial neural networks. A non-linear hyperbolic tangent was selected as the activation function and could be written as:

$$y_i = \tanh\left(\sum w_i . x_i\right)$$

where $y_i$ represents the output of the $i$th node (neuron), $x_i$ represents the $i$th input connection, and $w_i$ is the weight of $x_i$.

The CNN model was trained based on Keras[73] (V2.2.5) and TensorFlow[74] (V1.14.0) frameworks. We designed 1D convnets and followed the suggestions of François Chollet's suggestion in Deep Learning with Python[75] to regularize the model and to optimize the hyperparameters: (1) Adding Dropout layer to drop out a number of output features of the Dense layers in training process; (2) Try different architectures: add or remove layers; (3) Try to add "L1" or "L2" regularizer to make the distribution of weight values more regular; (4) Try different hyperparameters, such as the number of units per layer or the learning rate of the optimizer, to find out the optimal configuration.

Finally, the CNN model was composed of four convolutional layers, four pooling layers, and three fully connected layers. For convolutional layers, the "filters" parameters were 32, 64, 128, and 256, and the remaining parameters were set as "kernel_size=3, strides=1, padding='same', activation='relu', kernel_regularizer= l1(0.001)"; for pooling layers, the parameters were set as "pool_size=2, strides=2, padding='same'"; for fully connected layers, after 1024-way ReLU (Rectified Linear Unit) layers and 512-way ReLU layers, a two-way softmax layer was designed to return an array of two probability scores. Combined with the model, we constructed the gene prioritization function as follows:

$$\mathrm{Score}(g) = \frac{e^{z_2}}{\sum_{j=1}^{2} e^{z_j}} \times 100$$

where $z_j$ is the $j$th input for softmax layer, $z_1$ is the hidden layer value for negative class that were labeled as 0, and $z_2$ is the hidden layer value for positive class that were labeled as 1. Score($g$) is the gene score within the range of 0–100.

**Blast and primer.** The blast and primer functions were built from Sequence-Server[76] (V1.0.11) and Primer3[77] (V0.4.0), respectively, the core functions were retained, and the interface was adjusted to adapt to ISwine.

**Database implementation**. ISwine was built using tomcat (V8.0, web server) and MySQL (V5.7)/MongoDB (V3.4, database server). The website was designed and implemented using the Spring (V4.3.2) framework, which reduced the page refresh time to enhance the user experience. ISwine adopted the MVC design pattern based on SpringMVC (V4.3.2) and MyBatis (V3.2.8). Data visualization was implemented by an open source echarts package (https://github.com/apache/incubator-echarts). The website was tested in several popular web browsers, which included Internet Explorer, Google Chrome, and Firefox.

**Statistics and reproducibility**. Mann-Whitney test and hyper geometric test were implemented by using the wilcox.test() and phyper() function in R language. All relevant data can be found in the supplementary data or tables, and the statistical results can be reproduced based on this information.

**Reporting summary**. Further information on research design is available in the Nature Research Reporting Summary linked to this article.

## Data availability
The datasets analysed in this study were collected from the SRA, ENA, GEO, and PubMed repository. All the publicly available genome, transcriptome, and literature information are listed in Supplementary Data 1, 2, and 5. The source data for Fig. 2 are available at Supplementary Tables 6, 11, and 13, and Supplementary Data 13.

## Code availability
The scripts for model training are available on GitHub at https://github.com/xiaolei-lab/ISwine.

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

## Acknowledgements

We thank Thomas A. Gavin, Professor Emeritus, Cornell University, for help with editing this paper. This work was supported by the National Natural Science Foundation of China [31902156, 31702087, 31730089], the National Key Research and Development Program [2016YFD0101900], the National Swine Industry Technology System [CARS-35], the Natural Science Foundation of Hubei Province of China [2019CFC855], and the Fundamental Research Funds for the Central Universities of China [2020IVB025, 2662018JC033].

## Author contributions

S.H.Z., X.H.Y., and X.L.L. designed the study. Y.H.F. performed the data collection and analyses under the assistance and guidance from X.Y.L., M.Y., M.J.Z., J.Y.X., Z.S.T., L.W., D.Y., Y.F., L.L.Y., H.Y.W., and S.L.Z. Y.H.F. developed and evaluated the model under the assistance and guidance from X.H.Y., X.L.L., D.D.Z., F.D., Y.P.Z., H.H.Z., and W.H.X. Y.H.F., and X.L.L. wrote the paper.

## Competing interests

The authors declare no competing interests.
