## [Peer Review File · Communications Biology]

Reviewers' comments:

Reviewer #1 (Remarks to the Author):

The Manuscript presents an integrated multi-omics knowledgebase with an embedded CNN model for prioritizing genes in a non-model organism - swine. In addition, in order to make the result of this work accessible to users, the Authors developed Swine (<http://iswine.iomics.pro/>), an online tool where they incorporated almost all the published swine data.

The Authors have done a substantial work of particular interest to the scientific community by processing an impressive number of genomic and non-genomic data (Terabytes of data) and at the same time, they provide a highly usable tool (a web-based tool) to consult the results obtained.

The Manuscript is very convincing and the conclusions are adequately supported by the experiments conducted and by the results obtained. In detail, the methodology used to combine the various omic data is innovative and at the same time solid and valid.

Major issue:

Although the methodology used is adequate for the purpose, in some cases the level of detail with which it is provided is not sufficient for the Reader to reproduce the results. I refer in particular to the section dedicated to CNN, where it must be clearly expressed which data you have at the input and which at the output. In particular, the overall size of the dataset must be clarified, with which criteria the Authors have divided into training and test sets, the number of examples provided to the network for the training set and the dimensions of each example. This information is disseminated in the manuscript and explained at a high level.

Minor issues:

At lines 26-27: "the model precision was 72.9% and F1-Measure was 73.4%; this demonstrated a very good prediction performance." At this point, the reader cannot know if precision 72.9% and F1-Measure 73.4% is a very good performance. The Authors should add something like "compared to the state-of-the-art works which have precision and recall around TOT%".

At line 109: I don't know if it depends on the Authors or on how the files were collected during the submission process, however, I find that the way the various supplementary materials are called may confuse the Reader a little. For a while, I believed that Supplementary Data 1 was Supplementary Table 1. I would put all the figures in a pdf file as you have already done, and all the other tables (Table and Data) in excel files. Or you can keep the actual schema, but put as Supplementary Materials three files: a pdf file with the figures, an excel or pdf file with the tables and an excel file with the various Data 1, Data 2, Data 3 ... in different spreadsheets.

At line 185: Although all the details emerge in the method, I would add synthetically how many pairs you have in the training set and how many in the test set and how many for each class (1, 0)

At line 197: "The two additional models were two black box-based deep learning models, multi-layer perceptron (MLP) and Convolutional Neural Networks (CNN), which have the potential to mine the interactions among the features." Although some of the literature considers deep learning models as "black boxes", they apply specific mathematical transformations and precise operators to obtain a result. Diplomatically, do not write black boxes. I noticed "black boxes" here and there in the manuscript, the comment applies everywhere (e.g. lines 227, 371,...).

At line 582: to me, this part is avoidable. However it is correct, so if the Authors think it is useful to the reader, it can be kept it.

Reviewer #2 (Remarks to the Author):

The goal of manuscript "A gene prioritization method based on an integrated swine omics knowledgebase and a convolutional neural network model" is to provide a framework for integration of four types of omics data (Variation, Expression, WGCNA and QTNs/QTLs) for

prediction of causal and non-causal genes. The manuscript is broadly divided into two sections, one where the already published data is collected from various sources for storage in a database and one machine learning part that aims at prediction. The database part is fine, but the machine learning part is methodologically flawed at several places. My main objections are as follows:

1. The construction of the training and feature data is unclear and questionable. In the Methods section the authors state "QTGs were always proved additionally by experimental certification after statistical analysis." What does this mean? I cannot even guess the meaning of it. Moreover, they also say "Compared with most QTLs and QTNs that were only detected to be statistically associated with objective traits, QTGs were more reliable." This is a very problematic statement. How can QTGs be separated from QTLs and QTNs since the latter (QTLs and QTNs) are the ones that determine the former (QTGs)? This also means that you're using your QTX data twice, both as part of the input and as output. Regarding feature construction, what is the logic behind the use of the number of SNPs and indels as a genomic variation feature? That more variable parts of the genome are more likely to be associated with a certain trait? This makes no sense, especially not for selected traits. The same problem applies to the number of QTLs and QTNs, it is likely that this only reflects the how well-studied (i.e. number of studies) of a trait. It is the effects of the QTNs and QTLs that are interesting.

2. There are several passages in the manuscript where the authors claim to infer causality. In order to do this it is necessary to perform proper causal analysis, which is not the case in the manuscript. There is a large literature about causal inference (e.g. Causal Inference in Statistics - A Primer by J. Pearl 2016) and its application in genomics using Mendelian randomization (e.g. doi: 10.3389/fgene.2018.00238). You can only claim association in your manuscript (see for example <https://doi.org/10.1038/nmeth.3587>).

3. The descriptions of the machine learning methods are not clear. It is very strange that the LR performs best with no regularization (penalty='none') given that you have high dimensional data. I suspect that there are some errors in your data setup. How were the four data sets merged, just in one matrix? The design of the MPL should be described under that section. What kind of CNN kernels were used 1d, 2d or even 3d? How did you perform hyperparameter optimization? I doubt that it is possible to perform cross-validation and grid search over that many hyperparameters.

4. I think the authors miss the most important question. Does the integration of multi-omics data improve over no integration? It would be interesting to see a comparison of this.

Minor comments:

l 24-27: A model precision of 72.9% is not a very good prediction performance. I would say that it is at best indicative. However, the point is not so much the actual number, it is if it improves comparatively with other methods.

l 69-76: There are several examples of the use of deep learning in genomic prediction (e.g. doi.org/10.1186/s12711-020-00531-z, doi: 10.3389/fgene.2020.00025, doi.org/10.1111/jbg.12468).

Figure 1,3 and 4 overlap considerably. It would be enough with one figure that describes the framework.

Reviewer #3 (Remarks to the Author):

Review of the manuscript entitled: A gene prioritization method based on an integrated swine omics knowledgebase and a convolutional neural network model.

The manuscript describes the development of a method for gene prioritization based on current swine genomic/transcriptomic and published QTL, gene, GWAS studies data. I consider there is a lot of work under and very interesting for readers and users. The main contribution of the study is the generation of the ISwine database, very useful for all swine researchers, even more when swine variation/genome database updates are not currently supported.

I would recommend some English style correction (i.e.: ", but stops at the association level"), mainly in the Introduction section, as well as shorten this section.

Some suggestions to improve ISwine as user:

-to cluster RNA-Seq data by breed/regions (European/Asian/wildboar/etc) because differences are well documented as well as for genome variants.

-to add Official Gene Symbol (in addition to Gene ID) as searching criteria (for Integration) but for

other tools (as keyword to search).

-to review and recode some breeds/swine population (for instance Negro Iberico/Retinto/Iberian are the same breed, Iberian pig breed).

COMMSBIO-20-0703-T: Responses to Reviewers' Comments

Dear Reviewers,

Thank you for reviewing our manuscript (COMMSBIO-20-0703-T) entitled “A *gene prioritization method based on an integrated swine omics knowledgebase and a convolutional neural network model*”. We highly appreciate the time and effort that you have dedicated to providing your valuable feedback on our manuscript. We are grateful to your insightful comments and professional suggestions. We have been able to incorporate changes to reflect the suggestions, and we highlighted the changes within the manuscript. Below, we provide the point-by-point responses to your comments and concerns; Your comments are in black regular font followed by our responses in *blue italics*.

Responses to Reviewer #1:

(1) The Manuscript presents an integrated multi-omics knowledgebase with an embedded CNN model for prioritizing genes in a non-model organism - swine. In addition, in order to make the result of this work accessible to users, the Authors developed Swine (<http://iswine.iomics.pro/>), an online tool where they incorporated almost all the published swine data.

The Authors have done a substantial work of particular interest to the scientific community by processing an impressive number of genomic and non-genomic data (Terabytes of data) and at the same time, they provide a highly usable tool (a web-based tool) to consult the results obtained.

The Manuscript is very convincing and the conclusions are adequately supported by the experiments conducted and by the results obtained. In detail, the methodology used to combine the various omic data is innovative and at the same time solid and valid.

Response: *We sincerely thank you for your positive assessment, and we appreciate that you believe the integrated multi-omics knowledgebase will be of particular interest to the scientific community.*

Major issue:

(2) Although the methodology used is adequate for the purpose, in some cases the level of detail with which it is provided is not sufficient for the Reader to reproduce the results. I refer in particular to the section dedicated to CNN, where it must be clearly expressed which data you have at the input and which at the output. In particular, the overall size of the dataset must be clarified, with which criteria the Authors have divided into training and test sets, the number of examples provided to the network for the training set and the dimensions of each example. This information is disseminated in the manuscript and explained at a high level.

Response: *Thank you for your professional suggestions. We used the data that*

*composed of 842 samples with 14 features to train the model, and we randomly selected 80% of the samples as the training set and the remaining 20% as the test set. In the revised manuscript, we provided the details for “Training Set Preparation”, “Feature Generation”, and “Model Training and Gene Prioritization” in the Methods section. Readers can easily reproduce the results by using **supplementary data 6** and codes on GitHub at: <https://github.com/xiaolei-lab/ISwine>. (Line 536-544, Line 552-562, Line 566-575, Line 606-613)*

Minor issues:

(3) At lines 26-27: “the model precision was 72.9% and F1-Measure was 73.4%; this demonstrated a very good prediction performance.” At this point, the reader cannot know if precision 72.9% and F1-Measure 73.4% is a very good performance. The Authors should add something like “compared to the state-of-the-art works which have precision and recall around TOT%”.

Response: *Thank you for pointing out the issue. Compared with the recall of 64% and 79% in model organisms of Arabidopsis and rice¹, our study demonstrated a good prediction performance in a non-model organism -- swine. Although the recall in rice was higher than in swine, it only prioritized genes in the QTL regions, but we prioritized genes in the whole swine genome. We described and discussed this in the revised manuscript. (Line 26-27, Line 376-379)*

(4) At line 109: I don't know if it depends on the Authors or on how the files were collected during the submission process, however, I find that the way the various supplementary materials are called may confuse the Reader a little. For a while, I believed that Supplementary Data 1 was Supplementary Table 1. I would put all the figures in a pdf file as you have already done, and all the other tables (Table and Data) in excel files. Or you can keep the actual schema, but put as Supplementary Materials three files: a pdf file with the figures, an excel or pdf file with the tables and an excel file with the various Data 1, Data 2, Data 3 ... in different spreadsheets.

Response: *Thank you very much for your suggestions. According to the submission requirements and your suggestions, we have put the Supplementary Materials into two files, one is Supplementary Information (pdf file), which contains Supplementary Figures and Tables, and the other one is Supplementary Data (excel file), which contains various Supplementary Data in different spreadsheets.*

(5) At line 185: Although all the details emerge in the method, I would add synthetically how many pairs you have in the training set and how many in the test set and how many for each class (1, 0)

Response: *Many thanks for your suggestion. We used data comprised of 421 positive gene-trait pairs and 421 negative gene-trait pairs to train the model, and we randomly selected 80% of the pairs as the training set and the remaining 20% as the test set. We added the details as you have suggested, and we also provided more details in the revised manuscript. (Line 536-544, Line 552-554)*

(6) At line 197: “The two additional models were two black box-based deep learning models, multi-layer perceptron (MLP) and Convolutional Neural Networks (CNN),

which have the potential to mine the interactions among the features.” Although some of the literature considers deep learning models as "black boxes", they apply specific mathematical transformations and precise operators to obtain a result. Diplomatically, do not write black boxes. I noticed “black boxes” here and there in the manuscript, the comment applies everywhere (e.g. lines 227, 371,...).

Response: *Thank you for pointing out the issue. We totally agree with you. A number of methods and tools (e.g., the lime framework) could be conducted to understand the working principle of the deep learning models. In the revised manuscript, we changed our tone and avoided the description of ‘black boxes’ for deep learning models. (Line 191, Line 242,Line 389,Line 396)*

(7) At line 582: to me, this part is avoidable. However it is correct, so if the Authors think it is useful to the reader, it can be kept it.

Response: *Thank you for your suggestion. We agree with you that these contents are relatively basic. Considering that some readers only have the biological background, we moved this part to the Supplementary Information file.*

Reviewer #2 (Remarks to the Author):

The goal of manuscript "A gene prioritization method based on an integrated swine omics knowledgebase and a convolutional neural network model" is to provide a framework for intergration of four types of omics data (Variation, Expression, WGCNA and QTNs/QTLs) for prediction of causal and non-causal genes. The manuscript is broadly divided into two sections, one where the already published data is collected from various sources for storage in a database and one machine learning part that aims at prediction. The database part is fine, but the machine learning part is methodologically flawed at several places. My main objections are as follows:

Response: *Thank you for your positive comments to the database part. In the revised manuscript, we improved the understandability of the machine learning part and addressed your issues by adding new experiments and statements.*

(1) The construction of the training and feature data is unclear and questionable. In the Methods section the authors state "QTGs were always proved additionally by experimental certification after statistical analysis." What does this mean? I cannot even guess the meaning of it. Moreover, they also say "Compared with most QTLs and QTNs that were only detected to be statistically associated with objective traits, QTGs were more reliable." This is a very problematic statement. How can QTGs be separated from QTLs and QTNs since the latter (QTLs and QTNs) are the ones that determine the former (QTGs)? This also means that you're using your QTX data twice, both as part of the input and as output. Regarding feature construction, what is the logic behind the use of the number of SNPs and indels as a genomic variation feature? That more variable parts of the genome are more likely to be associated with a certain trait? This makes no sense, especially not for selected traits. The same problem applies to the number of QTLs and QTNs, it is likely that this only reflects the how

well-studied (i.e. number of studies) of a trait. It is the effects of the QTNs and QTLs that are interesting.

Response: *Thank you for pointing out the issue. The three abbreviations of QTLs, QTGs, and QTNs were defined as genomic regions, genes, and SNPs that were reported to be associated with any specific trait in previously published literatures. We apologize that the definitions and abbreviations were confusing in the former version. In the revised version, we changed QTLs, QTGs, and QTNs to QTALs, QTAGs, and QTANs, and the added letter 'A' represents 'associated'. (Line 158-160)*

To create a reliable dataset to train the machine learning model is very important, however, it is difficult to determine whether a gene is related to an objective trait, especially in non-model organism researches. In swine research, QTALs and QTANs were always identified by the statistical analyses only (e.g., QTL mapping or GWAS) without any experimental certification. Differently, the published studies for QTAGs were always conducted with an additional experimental certification after statistical analysis, and these QTAGs are more reliable compared with QTALs and QTANs. Therefore, we used 421 QTAG – trait pairs as positive samples. In our positive samples, 62.5% and 56.5% of the QTAGs did not overlap with QTANs and QTALs, respectively. This showed that although there was a certain relationship between QTAG and QTAL/QTAN, the QTALs/QTANs did not determine the QTAGs. On the other hand, 421 randomly assigned pairs of genes and traits were used as negative samples and we checked manually to confirm that the gene-trait pairs in negative samples were never published to be associated. Therefore, we believe that the quality of positive samples and negative samples is fine for model training. The details were provided in “Training Set Preparation” at Line 536-544.

*We generated a total of 14 features for training the gene prioritization model, which included 10 genomic features, two transcriptome features, and two literature features. **Genome features:** At the genomic level, we used the number of SNPs/indels within or nearby the gene as variation features for two reasons: (1) when SNPs/indels exist within a gene or in a regulatory region nearby a gene, they may play a direct role in regulating the gene function and, therefore, also the trait, especially for nonsynonymous variations. (2) the genes that do not occur within or nearby variations in a large population always have difficulty affecting a trait. **Transcriptome features:** At the transcriptome level, genes need to be expressed to perform their functions to affect the traits, and the strength of the gene functions is always related to the gene expression level. Therefore, the expression levels of the genes in the target tissue (s) were treated as an expression feature, and the correlation coefficient between the co-expression module and the tissue was used as a network feature. **Literature features:** Finally, consulting the literature is an important strategy to determine whether a gene is related to an objective trait. We agree with you that this feature may reflect how well-studied a QTAL/QTAN was for a trait. However, considering a large number of published studies, if a gene is reported to be associated with a trait a large number of times, it is more likely to be a reliable candidate gene. Therefore, we used the number of QTALs and QTANs, which overlapped with the gene region as the literature features. (Line 555-575)*

(2) There are several passages in the manuscript where the authors claim to infer causality. In order to do this it is necessary to perform proper causal analysis, which is not the case in the manuscript. There is a large literature about causal inference (e.g. Causal Inference in Statistics - A Primer by J. Pearl 2016) and its application in genomics using Mendelian randomization (e.g. doi: 10.3389/fgene.2018.00238). You can only claim association in your manuscript (see for example <https://doi.org/10.1038/nmeth.3587>).

Response: *Thank you very much for pointing out the issue. We agree with you that the ISwine can only recommend credible candidate genes from the candidate genes instead of causal genes. Following your suggestion, we changed the “causal genes” to “credible candidate genes” in the revised manuscript.*

(3) The descriptions of the machine learning methods are not clear. It is very strange that the LR performs best with no regularization (penalty='none') given that you have high dimensional data. I suspect that there are some errors in your data setup. How were the four data sets merged, just in one matrix? The design of the MPL should be described under that section. What kind of CNN kernels were used 1d, 2d or even 3d? How did you perform hyperparameter optimization? I doubt that it is possible to perform cross-validation and grid search over that many hyperparameters.

Response: *Thank you for your professional suggestion. We prepared 842 samples, generated 10 genomic features, two transcriptome features, and two literature features, and finally trained the model with this matrix with a dimension of 842 * 14. For the training of LR, SVC, and MLP models, we directly used the “LogisticRegression”, “LinearSVC”, and “MLPClassifier” functions in the scikit-learn framework without special design, respectively, and the appropriate parameters were also directly obtained by using the “GridSearchCV” function in the scikit-learn framework. We checked and updated the parameters of the LogisticRegression model and found that the model performed best without regularization (F1 = 64.94%), and when penalty = ‘l1’, its F1 also reached 64.84% (These F1 scores are “GridSearchCV” results and do not represent the score of the final model). In addition, by randomly reassigning the training set and the test set, the model will also appear to perform best with regularization. We infer that the regularization has less impact on the model performance under the current data.*

For the CNN model, we designed a one-dimensional convolutional neural network. We agree with you that it is difficult to grid search the hyperparameters, so we followed the suggestions of François Chollet in Deep Learning with Python² to regularize the model and to optimize the hyperparameters: 1) Adding Dropout layer to drop out a number of output features of the Dense layers in training process; 2) Try different architectures: add or remove layers; 3) Try to add ‘L1’ or ‘L2’ regularizer to make the distribution of weight values more regular; 4) Try different hyperparameters, such as the number of units per layer or the learning rate of the optimizer, to find out the optimal configuration.

To address your concerns, we provided all the details for “Training Set Preparation”, “Feature Generation”, and “Model Training and Gene Prioritization” in the revised manuscript and shared the codes at

<https://github.com/xiaolei-lab/ISwine>. Researchers can easily reproduce the results by using the supplementary data and codes. (Line 536-544, Line 552-562, Line 566-575, Line 606-613)

(4) I think the authors miss the most important question. Does the intergration of multi-omics data improve over no integration? It would be interesting to see a comparison of this.

Response: Thank you for your professional suggestion. We trained the models of LR, LinearSVC, MLP, and CNN by using the features of genome, transcriptome, and literature data. We found that among these models, the performances of models that trained by multi-omics information were better than that of single omics, and the methods based on neural network were superior to the linear methods (Supplementary Table 9). The results indicated that the performance of the model constructed using only genomic features was the best, followed by the transcriptome, and the literature was the worst. This was consistent with the evaluation results of lime, which indicated that the evaluation results of the integrated model were credible. In addition, we found that the performance of the model trained only with genomic features was closer to the integrated model, which may be because the genomic features accounted for 71.43% (10/14) of all features. However, from a biological point of view, the information of transcriptome and literature was more interpretable; all the transcriptome features and literature features were significantly different in the positive and negative datasets (Supplementary Table 10, Supplementary Figure 15), which also confirmed this; this suggested that some unknown regulatory mechanisms in the genomic layer needed to be mined. The integration of a number of features from multiple omics not only helped to improve the performance of the prediction model, but also enhanced the interpretability of the model. We have added this result in the revised manuscript. (Line 219-238)

Supplementary Table 9. Performances (F1- Measure) comparison of the integrated models and single omics models.

Omics	LinearSVC	SVC	MLP	CNN
genome	0.613	0.599	0.689	0.711
transcriptome	0.489	0.539	0.608	0.614
literature	0.367	0.367	0.460	0.347
multi-omics	0.623	0.612	0.701	0.730

The F1- Measure is used to measure the performance of the model, and the performance of multi-omics is better than that of single omics, and the method based on neural network is superior to the linear method. LR: Logistic regression; LinearSVC: Linear Support Vector Classifier; MLP: Multi-Layer Perceptron; CNN: Convolutional Neural Networks.

Supplementary Table 10. The mean values of 14 features in positive and negative samples.

Feature	Positive	Negative	P
Upstream_snp	57.33	60.44	3.89E-01
Downstream_snp	63.76	64.24	4.34E-01
Intron_snp	1,894.59	1,143.98	1.06E-14***
Synonymous_snp	48.13	25.38	5.88E-22***

Nonsynonymous_snp	71.03	46.24	7.26E-18***
Upstream_indel	3.82	3.84	9.06E-01
Downstream_indel	3.78	4.01	4.60E-01
Intron_indel	121.86	76.90	2.67E-15***
Synonymous_indel	4.15	2.69	2.47E-14***
Nonsynonymous_indel	0.11	0.12	8.43E-01
Module	0.25	0.13	1.02E-22***
Expression	3.49	1.60	1.53E-36***
QTN	5.65	0.69	1.98E-12***
QTL	1.66	0.59	2.12E-07***

Nine of the 14 features showed significant differences between two datasets. ***: $P < 0.01$.

Supplementary Figure 15. The comparison of 14 features in positive and negative samples. Nine of the 14 features showed significant differences between positive and negative samples. The statistical significance was calculated by the Mann-Whitney test. ***: $P < 0.01$.

Minor comments:

(5) 124-27: A model precision of 72.9% is not a very good prediction performance. I would say that it is at best indicative. However, the point is not so much the actual number, it is if it improves comparatively with other methods.

Response: *Thank you for pointing out the issue. Compared with the recall of 64% and 79% in model organisms of Arabidopsis and rice¹, our study demonstrated a good prediction performance in a non-model organism -- swine. Although the recall in rice was higher than in swine, it only prioritized genes in the QTL regions, but we prioritized genes in the whole swine genome. We described and discussed this in the revised manuscript (Line 26-27, Line 376-379).*

(6) 169-76: There are several examples of the use of deep learning in genomic prediction (e.g. doi.org/10.1186/s12711-020-00531-z, doi:10.3389/fgene.2020.00025, doi.org/10.1111/jbg.12468).

Response: *Many thanks for your suggestion. We added the appropriate citations in the revised manuscript. (Line 68)*

(7) Figure 1,3 and 4 overlap considerably. It would be enough with one figure that describes the framework.

Response: *Thank you very much for pointing out the issue. We have merged Figure 3 and Figure 4 into one figure and deleted the information that was repeated with Figure 1. In the revised manuscript, Figure 1 shows the schematic of the gene prioritization framework, and Figure 3 shows the structure and interface of the database. (Line 331)*

Reviewer #3 (Remarks to the Author):

Review of the manuscript entitled: A gene prioritization method based on an integrated swine omics knowledgebase and a convolutional neural network model.

(1) The manuscript describes the development of a method for gene prioritization based on current swine genomic/transcriptome and published QTL, gene, GWAS studies data. I consider there is a lot of work under and very interesting for readers and users. The main contribution of the study is the generation of the ISwine database, very useful for all swine researchers, even more when swine variation/genome database updates are not currently supported.

Response: *We sincerely thank you for your positive assessments, and we appreciate that you believe our knowledgebase will be of great interest to the research community.*

(2) I would recommend some English style correction (i.e.: “, but stops at the association level”), mainly in the Introduction section, as well as shorten this section.

Response: *Thank you for pointing out the issue. In the revised version, we corrected the English style problems (i.e.: “but stops at the association level” was changed to*

*“but these analyses always stop at the “association” level”, **Line 46**), simplified the Introduction section, and improved the English writing of the manuscript with help of Thomas A. Gavin, Professor Emeritus, Cornell University.*

Some suggestions to improve ISwine as user:

(3) -to cluster RNA-Seq data by breed/regions (European/Asian/wildboar/etc) because differences are well documented as well as for genome variants.

Response: *Thank you for your professional suggestion, and it is a very valuable attempt. However, all the RNA-seq data in this study were derived from public data, and it was difficult to obtain the breed/regions information for most samples. Considering that RNA-seq data are always used for the study of the temporal and spatial specificity of gene expression, we clustered the RNA-seq data by using the tissue information.*

(4) -to add Official Gene Symbol (in addition to Gene ID) as searching criteria (for Integration) but for other tools (as keyword to search).

Response: *Many thanks for your suggestion. We added this function in ISwine, and users can search Integration, Variation, Expression, and QTX databases by using Official Gene Symbol now.*

(5) -to review and recode some breeds/swine population (for instance Negro Iberico/Retinto/Iberian are the same breed, Iberian pig breed).

Response: *Thank you for pointing out the issue. We checked the breed information of these samples and recoded the Negro Iberico breed and Retinto breed to Iberian breed following your suggestion.*

We sincerely thank the **reviewer #1**, **reviewer #2**, and **reviewer #3** again for the kindly criticizes and professional comments. We look forward to hearing from you regarding our submission and to respond to any further questions and comments you may have.

Reference

- 1 Lin, F., Fan, J. & Rhee, S. Y. QTG-Finder: A Machine-Learning Based Algorithm To Prioritize Causal Genes of Quantitative Trait Loci in Arabidopsis and Rice. *G3 (Bethesda)* **9**, 3129-3138, doi:10.1534/g3.119.400319 (2019).
- 2 Chollet, F. *Deep Learning with Python*. (Manning Publications, 2017).

REVIEWERS' COMMENTS:

Reviewer #1 (Remarks to the Author):

The Reviewer thanks the Authors for the precious work done. The Authors resolved all the critical points that emerged from the previous review.

Reviewer #2 (Remarks to the Author):

The authors have responded to the comments and revised the manuscript satisfactorily.

Reviewer #3 (Remarks to the Author):

The authors have followed my recommendations to improve the manuscript. English style corrections, recodification of some breeds/swine population have been conducted. I would really appreciate clustering RNA-Seq data conditional on breed or region, but I understand this information is not always available. I suggest to reduce discussion by deleting some redundant paragraphs in lines 340-352 and 416-425.

COMMSBIO-20-0703-T: Responses to Reviewers' Comments

Dear Reviewers,

Thank you again for reviewing our manuscript (COMMSBIO-20-0703-T) entitled “A gene prioritization method based on a swine multi-omics knowledgebase and a deep learning model”. We appreciate all reviewers for their recommendations of publication. We also appreciate Reviewer #3 for the suggestions. We have revised our manuscript by following the suggestions exactly. Below, we provide the point-by-point responses to your comments and concerns; Your comments are in black regular font followed by our responses in *blue italics*.

Responses to Reviewer #1:

(1) The Reviewer thanks the Authors for the precious work done. The Authors resolved all the critical points that emerged from the previous review.

Response: *Thank you for reviewing our manuscript. Your comments helped us to improve our manuscript.*

Responses to Reviewer #2:

(1) The authors have responded to the comments and revised the manuscript satisfactorily.

Response: *Thank you for reviewing our manuscript. Your comments helped us to improve our manuscript.*

Responses to Reviewer #3:

(1) The authors have followed my recommendations to improve the manuscript. English style corrections, recodification of some breeds/swine population have been conducted. I would really appreciate clustering RNA-Seq data conditional on breed or region, but I understand this information is not always available.

Response: *We sincerely thank you for your previous help to us to improve the manuscript. In the follow-up maintenance of ISwine, we will try our best to improve this information.*

(2) I suggest to reduce discussion by deleting some redundant paragraphs in lines 340-352 and 416-425.

Response: *Thank you very much for pointing out the issue. Following your suggestion, we removed the redundant information in the revised manuscript.*